# Unregulated *GmAGL82* Due to Phosphorus Deficiency Positively Regulates Root Nodule Growth in Soybean

**DOI:** 10.3390/ijms25031802

**Published:** 2024-02-01

**Authors:** Jia Song, Ying Liu, Wangxiao Cai, Silin Zhou, Xi Fan, Hanqiao Hu, Lei Ren, Yingbin Xue

**Affiliations:** 1College of Coastal Agricultural Science, Guangdong Ocean University, Zhanjiang 524088, China; s929497809@163.com (J.S.); liuying85168@gdou.edu.cn (Y.L.); huhanqiao@gdou.edu.cn (H.H.); 2South China Branch of National Saline-Alkali Tolerant Rice Technology Innovation Center, Zhanjiang 524088, China; 3College of Chemistry and Environment, Guangdong Ocean University, Zhanjiang 524088, China; m13715805122@163.com (W.C.); zsl15915153127@163.com (S.Z.); f18824902430@163.com (X.F.)

**Keywords:** soybean, *GmAGL82*, nodule, phosphorus deficiency, RNA-seq

## Abstract

Nitrogen fixation, occurring through the symbiotic relationship between legumes and rhizobia in root nodules, is crucial in sustainable agriculture. Nodulation and soybean production are influenced by low levels of phosphorus stress. In this study, we discovered a MADS transcription factor, *GmAGL82*, which is preferentially expressed in nodules and displays significantly increased expression under conditions of phosphate (Pi) deficiency. The overexpression of *GmAGL82* in composite transgenic plants resulted in an increased number of nodules, higher fresh weight, and enhanced soluble Pi concentration, which subsequently increased the nitrogen content, phosphorus content, and overall growth of soybean plants. Additionally, transcriptome analysis revealed that the overexpression of *GmAGL82* significantly upregulated the expression of genes associated with nodule growth, such as *GmENOD100*, *GmHSP17.1*, *GmHSP17.9*, *GmSPX5*, and *GmPIN9d*. Based on these findings, we concluded that *GmAGL82* likely participates in the phosphorus signaling pathway and positively regulates nodulation in soybeans. The findings of this research may lay the theoretical groundwork for further studies and candidate gene resources for the genetic improvement of nutrient-efficient soybean varieties in acidic soils.

## 1. Introduction

Soybean (*Glycine max* [L.] Merr) is not only rich in protein, fat, vitamins, and trace elements, such as calcium, iron, and magnesium, but also constitutes an important global food and oil crop, fertilizer, and high-quality feed for livestock [1]. Soybean possesses the ability to form root nodules through a symbiotic association with nitrogen-fixing rhizobia present in the soil. This symbiotic association is viewed as the most efficient mode of biological nitrogen fixation, being able to convert atmospheric nitrogen (N_2_) into a readily soluble and nontoxic form, primarily ammonium (NH_4_^+^), which is subsequently utilized by plant cells to synthesize a wide range of biomolecules [2,3]. Reduced nitrogen fertilizer application or inoculation with rhizobia at lower nitrogen levels can promote the formation of more nodules in soybeans, increasing soybean yield by over 90% [4]. This demonstrates the significant contribution of biological nitrogen fixation in soybean cultivation to increased yields and reduced nitrogen fertilizer usage [5]. In sustainable agricultural production, intercropping with soybeans not only enhances the yield of food crops but also improves soil fertility and field ecological conditions, thereby regulating carbon and nitrogen dynamics and phosphorus efficiency [6,7].

Upwards of 50% of the earth’s potentially cultivable lands consist of acidic soils, posing limitations to crop productivity in these regions [8]. The soils in the southern regions are predominantly characterized by acidic red soils and are subject to high temperatures and abundant rainfall throughout the year. Due to leaching, resulting in the loss of nutrients and alkaline solutes, the soil ultimately experiences acidification, which is further exacerbated by the release of protons (H^+^) through the conversion reactions of various carbon, nitrogen, and sulfur compounds in agricultural soils [9]. Crops grown on acidic soils often face stress from low pH, aluminum toxicity, and low phosphorus stress [10,11,12]. The effectiveness of essential nutrients [phosphorus (P), molybdenum (Mo), calcium (Ca), and magnesium (Mg)] is diminished due to the acidity of the soil, resulting in detrimental effects on plant growth and development [13,14]. Despite legumes’ ability to form nitrogen-fixing symbiotic relationships with soil microorganisms (rhizobia), making them promising test crops for acidic soils, soybeans can still be directly or indirectly damaged in acidic soil [15]. Acidic soils are unfavorable for plant growth due to their detrimental effects on plant roots and reduced accessibility to essential nutrients, resulting in a significant decline in overall plant growth [16,17]. Low-pH soils have detrimental effects on the symbiotic relationship between legume crops and rhizobia, notably impeding nodulation and subsequently reducing nitrogen fixation [9]. Furthermore, acidic soils hinder both the growth of legume plants and the development of nodules, as high levels of aluminum and iron toxicity impair the formation and function of nodules [18,19]. Based on reports, phosphorus-efficient soybean genotypes exhibit better adaptability to acidic soil conditions than phosphorus-inefficient genotypes, leading to increased biomass and longer root length, particularly in the presence of sufficient phosphorus supply [11].

Phosphorus is a crucial nutrient for plant growth and development. However, the presence of insoluble organic phosphates, resulting from the combination of metal ions and phosphorus in acidic soils, makes it challenging for plants to absorb and utilize phosphorus, significantly limiting crop productivity [20,21,22]. Reports suggest that at least 30–40% of global crop yields are negatively affected by phosphorus deficiency [23,24]. Research indicates that the N_2_ fixation of legumes demands greater amounts of P for optimal functioning than non-nodular plants; this increased phosphorus requirement may be associated with this element’s essential role in facilitating vital energetic transformations within the nodules [25]. Moreover, the metabolic processes of symbiotic nitrogen-fixing legumes require a substantial amount of phosphorus, and deficiencies in this regard lead to ammonia assimilation into amino acids and ureides, resulting in an insufficient energy supply for plant development [26,27]. Phosphorus deficiency severely hinders growth metabolism, biomass production, the quality of plants, and the nitrogen fixation capacity of nodules [28,29]. Therefore, studying the regulatory mechanisms of phosphorus nutrition in legume crop nodules and developing phosphorus-efficient genetic resources hold significant market value in terms of achieving high crop yields, improved quality, and genetic enhancements.

In response to long-term phosphorus deficiency, plants have evolved adaptive mechanisms to cope with low-phosphorus stress, such as changing their root architecture, enhancing the expression of high-affinity phosphate transporter genes, increasing the activity of acid phosphatases, etc. [30,31,32,33,34]. In the present era, a significant body of research has revealed that plants regulate root morphology in response to the perceived availability and distribution of soil phosphorus, thereby enhancing the efficiency of phosphorus uptake by the root system [31,35,36]. In soybeans, *GmEXPB2* has been identified as an important gene involved in controlling the morphological architecture of the root system [37]. In addition to the root morphological architecture described above, high-affinity phosphorus transporter expression is also increased in response to low-phosphorus stress. Plants employ a synergistic interplay of various phosphate transporters to facilitate the uptake and transport of phosphorus, especially the high-affinity phosphorus transporter family, which includes five categories, namely, PHT1, PHT2, PHT3, PHT4, and PHT5 [38]. The increased activity of acid phosphatase serves as a biochemical indicator of phosphorus deficiency in plants [39,40]. Multiple purple acid phosphatases (PAPs) are involved in the activation of exogenous organic phosphorus [41]. Moreover, symbiosis with rhizobia or mycorrhizal fungi is also an important mechanism for plant adaptation to low-phosphorus stress. It has been noted that plants can acquire and utilize phosphorus through interactions with mycorrhizal fungi. Under conditions of low-phosphorus stress, the majority of terrestrial plants, with the exception of several, such as *Arabidopsis*, can establish a symbiotic relationship with mycorrhizal fungi [42,43]. This symbiotic mechanism involves enhancing phosphorus uptake by expanding extra-root mycelium, inducing the expression of phosphorus transporters, and promoting the activation of exogenous phosphorus by the symbiotic plants [44]. Recent studies have demonstrated that the inoculation of legumes with rhizobia not only enables nitrogen fixation but also enhances the uptake of exogenous phosphorus. It has been observed that soybean inoculation with rhizobia relies on the secretion of more H^+^ in order to absorb and utilize insoluble phosphorus [45]. However, the specific regulatory mechanism behind this process remains in need of study.

A prominent family of plant proteins dubbed the MADS-box transcription factor, are widely distributed in fungi, plants, and animals [46]. The MADS-box transcription factor derives its name from four representative genes: MCM1 (minichromosome maintenance 1) found in *Saccharomyces cerevisiae*, AG (agamous) identified in *Arabidopsis thaliana*, DEF (deficiens) present in *Antirrhinum majus*, and SRF (serum response factor) observed in *Homo sapiens*. These genes collectively contribute to the diverse distribution of the MADS-box family across various organisms [47]. The N-terminal region of the encoded protein contains a highly conserved MADS DNA-binding domain (M), which possesses the ability to bind to specific DNA sequences known as CArG boxes [CC(A/T)_6_GG] through a consensus recognition mechanism [48]. MADS-box genes in plants can be grouped into two clades based on their protein structure: Type I and Type II. Genes belonging to Type I exhibit an M-domain in their protein structure. The protein structure of Type II MADS-box genes, which are sometimes referred to as MICK-type genes, consists of a sequence of domains including the M-domain, followed by an intervening domain (I), then a keratin-like domain (K), and finally a C-terminal domain (C) [49,50,51,52]. Generally speaking, the structure of Type I MADS genes is characterized by a simple arrangement of 1–2 exons, and there is a scarcity of studies reporting their functions, which appear to be associated with plant reproduction alone [53,54,55]. Type II MADS genes are unique to plants and are currently the most studied variety of relevant genes. The most renowned among this group are those involved in floral organ formation and development [56,57]. MADS-box transcription factors have been found to play crucial roles in controlling several aspects of plant development, including floral organ formation, flowering time, root growth, fruit development, and ripening [58,59,60]. Furthermore, several studies assert that certain MADS family transcription factors can modulate plant reactions to various abiotic stress conditions, such as drought, cold, and salt stress [61,62,63,64,65,66]. In soybeans, studies have discovered that the MADS gene *GmNMHC5*, which is regulated by sugar, plays a crucial role in controlling lateral root development and the formation of root nodules [65]. *GmNMH7* is likely to be involved in the nod factor (NF) signaling pathway and may play a negative regulatory function in nodulation, potentially by modulating the levels of GA3 [67].

Within this research, we identified a MADS family member, Glyma19G045900, that showed enhanced expression in root nodules under low-phosphate conditions. We named it *GmAGL82.* Exploring the overexpression of *GmAGL82* in soybean composite plants, we investigated the role of *GmAGL82* in root nodules, examining its effects on nitrogen and phosphate levels as well as the number of root nodules. Furthermore, via transcriptome sequencing, we found that the overexpression of *GmAGL82* exerts an influence on crucial genes associated with the positive regulation of significant functions such as soybean nodulation and growth development, phenomena that have been previously documented. This discovery expands our understanding of the physiological mechanisms and molecular regulatory pathways through which *GmAGL82* is involved in regulating soybean nodulation under low-phosphate stress, contributing to our understanding of the specific functions of this gene in facilitating soybean nodulation under low-phosphate stress. Of additional interest is the role of this member in improving soybean quality. This knowledge can be valuable in the genetic engineering of soybeans by providing alternative genes for selection to develop varieties that are high-yielding, high-quality, and efficient with regard to nitrogen–phosphorus utilization in acidic soils.

## 2. Results

### 2.1. Evolutionary Tree Analysis of MADS Family Proteins and Conserved Protein Motif Analysis

A phylogenetic analysis was conducted, comparing the protein sequence of the *GmAGL82* gene with other well-established MADS family members in model plants, such as *Arabidopsis thaliana*, rice (*Oryza sativa*), and soybean, in order to elucidate the evolutionary role of the *GmAGL82* gene (Figure 1). After the evolutionary tree analysis, we found that these MADS members exhibit a discernible segregation into two distinct groups, namely, Type I and Type II (Figure 1). Among them, *GmAGL82* belongs to Type I and exhibits the closest homology to *OsMADS87* in rice. Members such as *AtAGL6/15/16/17/21/31* and *AtANR1* in *Arabidopsis thaliana*, *OsMADS26* in rice, and *GmAGL1/9/11/15*, *GmNMH7*, *GmNMHC5*, *GmMADS28*, and *GmSEP1* in soybean all belong to Type II (Figure 1).

Furthermore, the conserved motif analysis of MADS family proteins was performed using the online tool MEME. The results showed that only one conserved motif, Motif1, was identified in the protein sequences of GmAGL82 and OsMADS87 (Appendix A). In line with the phylogenetic tree analysis, it was observed that the AtSEP2 protein sequence within the remaining MADS Type I only contained Motif3 (Appendix A). Conversely, the proteins OsMADS78, OsMADS79, AtAGL23, and AtAGL62 shared both Motif1 and Motif3 (Appendix A). This indicates that these two motifs may contribute to the conservation of protein function within the MADS Type I. Except for OsMADS62, MADS Type II proteins exhibited a relatively high number of motifs, ranging from 3 to 7 (Appendix A). An analysis of the structural characteristics of each conserved motif revealed that Motif1 was the most prevalent motif within the MADS gene family, appearing in all 50 genes. Members within the same evolutionary branch displayed similar or comparable motif distributions, implying shared or similar functionality (Appendix A).

### 2.2. Analysis of Cis-Acting Elements of the GmAGL82 Gene Promoter

The promoter region of the *GmAGL82* gene, consisting of a 2000 bp sequence upstream of the ATG codon, was selected. We then analyzed its cis-elements. The *GmAGL82* gene promoter contained 12 crucial response elements related to light, plant hormones, anaerobic conditions, and MYB transcription factors (Appendix A). Among them, there were five light-responsive elements, including two Box 4 elements, two G-box elements, and one ATCT element (Appendix A). In the plant hormone category, there were four elements, including two CGTCA elements involved in the methyl jasmonate response, one ABBRE element for the abscisic acid response, and a GARE element implicated in the gibberellin response (Appendix A). Additionally, there was one anaerobic-responsive element (ARE), one MYBHv1-binding site (CCAAT-box), and one MYB-binding site (MBSI) involved in the regulation of flavonoid biosynthesis genes (Appendix A).

### 2.3. Expression Pattern of GmAGL82

The expression pattern of *GmAGL82* was analyzed at different nodule growth stages under conditions of normal phosphorus supply (250 μM KH_2_PO_4_, +P) and phosphorus deficiency (5 μM KH_2_PO_4_, −P), as illustrated in Figure 2A. Nodules were harvested after 15, 20, 25, 40, 50, and 60 days under the respective phosphate treatments. During this period, the expression level of *GmAGL82* increased progressively, reaching its highest point on day 25, where it was 8.5 times higher than the figure for day 15 (Figure 2A). Subsequently, the nodules entered a senescence phase from day 50 onwards, accompanied by a reduction in *GmAGL82* expression. Significantly higher expression levels of *GmAGL82* were observed in the −P nodules on days 15, 20, and 25, with fold increases of 182.4, 20.3, and 2.4, respectively, compared to the +P nodules (Figure 2A).

The qRT-PCR analysis was employed to determine its expression levels in the leaves, roots, flowers, stems, capsules, seeds, and nodules of soybean following 40 days of growth (Figure 2B). The results revealed that *GmAGL82* gene expression could be detected in all examined tissues. Notably, *GmAGL82* exhibited higher expression levels in the flowers and nodules, with the highest expression observed in the nodules (Figure 2B). In contrast, the expression level was relatively lower in the roots, stems, leaves, capsules, and seeds, with the lowest expression observed in the seeds (Figure 2B). Compared to its expression in flowers, roots, stems, leaves, capsules, and seeds, the presence of *GmAGL82* increased to a remarkable extent in the nodules, with a level approximately 9.7 and over 100 times higher than that in flowers and the other tissues, respectively (Figure 2B).

### 2.4. GmAGL82 Subcellular Localization

To investigate the subcellular localization of the *GmAGL82* gene, this study utilized the transient transformation of tobacco leaves with *Agrobacterium tumefaciens* GV3101 carrying a *GmAGL82*-GFP fusion construct (*35S:GmAGL82-GFP*). Tobacco leaves transformed using the empty vector containing the GFP gene (*35S:GFP*) were used as a control (Figure 3). As shown in Figure 3, leaves transformed with the GFP empty vector exhibited strong green fluorescence in the nucleus, cytoplasm, and cell membrane. In contrast, leaves transformed with the *GmAGL82* gene primarily displayed strong green fluorescence in the nucleus, with weak fluorescence observed in the cell membrane (Figure 3). These results provide evidence that the *GmAGL82* gene is principally localized in the nucleus.

### 2.5. Effects of Overexpressing GmAGL82 on Soybean and Nodules Biomass

To investigate the function of the *GmAGL82* gene, this study constructed an overexpression vector, *GmAGL82-pTF101s*, and used the *Agrobacterium*-mediated hypocotyl transformation method to generate transgenic soybean plants overexpressing *GmAGL82*. These transgenic plants were then subjected to inoculation with rhizobia in order to analyze the impact of *GmAGL82* overexpression on the growth of soybean composite plants and nodules. The transgenic lines displaying *GmAGL82* overexpression, along with control lines transformed with an empty vector, were cultured in soybean nutrient solution for 28 days, and their phenotypes were observed (Figure 4A). As shown in the figure, the *GmAGL82* overexpression lines exhibited a trend of promoting nodule growth and increasing the number of nodules compared to the control lines (Figure 4A). The *GmAGL82*-overexpressing lines displayed 35% higher root biomass, 30% higher shoot biomass, and 33% higher whole-plant dry weight compared to the control (Figure 4B). Specifically, the *GmAGL82* overexpression lines demonstrated a 2.1-fold increase in nodule fresh weight (Figure 4C) and a 2.5-fold increase in the number of nodules (Figure 4D). A noteworthy 119% increase in the nodule count per unit root length was also observed (Figure 4E).

Although the overexpression of *GmAGL82* did not result in an increase in the phosphorus content within the root system, it significantly elevated the levels of phosphorus in the shoot, nodules, and the whole plant. Compared to the control, the phosphorus content increased by 29%, 133%, and 30% in the aboveground part, nodules, and the whole plant, respectively (Figure 4G). Simultaneously, a significant increase in nitrogen content was achieved through the overexpression of *GmAGL82*. In comparison to the control group, the nitrogen content in the shoot, root system, nodules, and whole plant increased by 77%, 44%, 44%, and 46%, respectively (Figure 4F). In comparison to the control, the overexpression of *GmAGL82* in the transgenic composite plants resulted in a 25% boost in soluble phosphorus in the nodules, while the soluble phosphorus concentration in the root hairs decreased by 23% (Figure 4H).

### 2.6. Statistics of Differentially Expressed Genes in the Root Systems and Root Nodules of the Control and GmAGL82-Overexpressing Plants

To investigate the regulatory network of root and nodule development mediated by the overexpression of *GmAGL82* in soybeans, we performed RNA sequencing (RNA-seq) analysis. By comparing the transcriptomes of root and nodule tissues between *GmAGL82*-overexpressing soybean plants (OX) and the control transformed using an empty vector (CK), we analyzed the global changes in gene expression (Figure 5) and explored the molecular processes regulated by the overexpression of *GmAGL82*. The criteria for selecting differentially expressed genes (DEGs) were as follows: gene expression changes exceeding a 2-fold threshold with a corrected *p*-value of less than 0.05. A total of 47,856 genes were identified in the soybean roots, among which 3309 genes showed differential expression between OX and CK, including 1924 upregulated genes and 1385 downregulated genes (Figure 5A, Appendix A). A total of 48,047 genes were identified in the nodules, with 3363 genes showing expression changes due to *GmAGL82* overexpression (Figure 5B, Appendix A). *GmAGL82* overexpression significantly enhanced the expression of 1400 genes, while repressing its expression for 1963 genes (Figure 5B, Appendix A). Furthermore, by comparing the DEGs between the roots and nodules, we found that there were 577 DEGs upregulated and 303 DEGs downregulated in both roots and nodules (Figure 5C). Additionally, 1331 and 817 DEGs underwent upregulation alone via the overexpression of *GmAGL82* in roots and nodules, respectively (Figure 5C). Further, only 1076 and 303 DEGs were downregulated due to the overexpression of *GmAGL82* in roots and nodules, respectively (Figure 5C). A total of 16 DEGs were upregulated in roots but downregulated in nodules, while 6 DEGs were downregulated in roots but consequently upregulated in nodules (Figure 5C).

### 2.7. Gene Ontology Analysis of Differentially Expressed Genes

To better comprehend the regulatory network of *GmAGL82* in response to low-phosphorus stress in soybean nodules, Gene Ontology (GO) enrichment analysis on the DEGs in both roots (OX vs. CK) and nodules (OX vs. CK) was conducted. We observed a remarkable enrichment in various biological processes, including the response to chemical stimuli and metabolic processes associated with small molecules, in the DEGs of the roots (Figure 6A). In terms of cellular component classification, the DEGs were predominantly enriched in the extracellular region (Figure 6A). The most enriched DEG molecular function was catalytic activity (Figure 6A). As with roots, the most enriched molecular function among the DEGs was also catalytic activity in nodules (Figure 6B). Further, the extracellular region was the DEGs predominantly enriched in nodules in terms of cellular component classification (Figure 6B). It was observed that the nodule DEGs underwent a notable enrichment for biological pathways linked to organic the substance catabolic process, carbohydrate metabolic process, and hormone metabolic process (Figure 6B).

### 2.8. Analysis of Differentially Expressed Genes Using Kyoto Encyclopedia of Genes and Genomes

By performing the Kyoto Encyclopedia of Genes and Genomes (KEGG) analysis, we compared the metabolic processes in which the DEGs are involved. Twenty metabolic processes were found to be primarily associated with DEGs in the roots. They were prominently enriched in terms of secondary metabolism, including 34 genes participating in phenylpropanoid biosynthesis, 5 genes associated with nitrogen metabolism, and 8 genes implicated in taurine and hypotaurine metabolism (Figure 7A). Additionally, they were involved in carbon metabolism, with four genes participating in glycolysis/gluconeogenesis and four genes involved in starch and sucrose metabolism (Figure 7A). Furthermore, we discovered an association with lipid metabolism, with six genes involved in fatty acid biosynthesis, six genes participating in the biosynthesis of unsaturated fatty acids, four genes implicated in fat digestion and absorption, and eight genes participating in alpha-linolenic acid metabolism (Figure 7A). They were also involved in amino acid metabolism, including six genes participating in valine, leucine, and isoleucine degradation, and eight genes with roles in pyrimidine metabolism (Figure 7A). Other metabolic pathways included proximal tubule bicarbonate reclamation; protein processing in the endoplasmic reticulum; zeatin, cutin, wax, suberine, and isoquinoline alkaloid biosynthesis; chloroalkane and chloroalkene degradation; butanoate metabolism; and the HIF-1, glucagon, and PPAR signaling pathways (Figure 7A).

In the nodules, we found that carbon metabolism was predominant and displayed a higher incidence (Figure 7B). Specifically, within this metabolic pathway, a total of 17 genes were identified as being correlated with starch and sucrose metabolism, while 7 genes participated in pentose and glucuronate interconversions, and an additional 6 genes were associated with galactose metabolism (Figure 7B). Secondary metabolism was also enriched, including 39 genes involved in phenylpropanoid biosynthesis and 3 participating in sesquiterpenoid and triterpenoid biosynthesis (Figure 7B). There was an enrichment in DEGs involved in amino acid metabolism, including 10 genes involved in valine, leucine, and isoleucine degradation; 7 genes involved in tryptophan metabolism; and 10 genes involved in purine metabolism (Figure 7B). Lipid metabolism was represented by four genes involved in fat digestion and absorption, and four genes involved in inositol phosphate metabolism (Figure 7B). Other enriched metabolic processes included the PPAR signaling pathway; protein processing in the endoplasmic reticulum; suberine, cutin, and wax biosynthesis; photosynthesis-antenna proteins; isoquinoline alkaloid biosynthesis; novobiocin biosynthesis; *Escherichia coli* biofilm formation; biosynthesis of ansamycins; brassinosteroid biosynthesis; sulfur metabolism; cyanoamino acid metabolism; chloroalkane and chloroalkene degradation; antigen processing and presentation; prodigiosin biosynthesis; neurotrophin signaling pathway; and thiamine metabolism (Figure 7B). These findings suggest that multiple metabolic pathways in soybeans may be regulated by *GmAGL82*.

### 2.9. Analysis of Differentially Expressed Genes Associated with Nodules

Through the analysis of DEGs, we identified several recently reported genes related to soybean nodulation, such as the sucrose synthase gene *GmENOD100*, the small heat shock proteins GmHSP17.1 and GmHSP17.9, the auxin transport-related gene *GmPIN9d*, etc. (Table 1). For example, overexpressing *GmAGL82* significantly upregulated the expression of *GmHSP17.1* and *GmHSP17.9* in both roots and nodules (Table 1). *GmAGL82* overexpression also upregulated the expressions of *GmENOD100*, *GmPIN8b*, *GmPIN9d*, *GmSPX5*, *GmNSP2b*, *GmPAP27e*, and *GmPHR2* in roots but not in nodules (Table 1). Meanwhile, *GmNAC181*, *GmNSP1a*, and *GmWRKY45* were significantly upregulated by the overexpression of *GmAGL82* in nodules. However, this was not the case in roots (Table 1). For the phosphate transporter genes, *GmAGL82* overexpression decreased the expressions of *GmPHT1.12* and *GmPHT3.1* in roots but increased the expressions of *GmPT11* and *GmPHO1.12* in roots, and *GmPHT4.7* in nodules (Table 1).

### 2.10. Quantitative Real-Time Polymerase Chain Reaction Validation of the Sequencing Results

To validate the results of transcriptome sequencing, 20 selected DEGs were used to perform the quantitative real-time polymerase chain reaction (qRT-PCR) analysis in roots (Figure 8A). The selected genes included 4 genes involved in phosphate transport, 10 genes related to soybean nodulation, and 6 genes that function as transcription factors. In the *GmAGL82* overexpression roots, the 19 candidate genes displayed a notable elevation in their expression levels when compared to the control group (CK). Among them, the high-affinity phosphate transporter gene *GmPT11* (Glyma.19G164300) displayed greater-than-16-fold upregulation in response to *GmAGL82* overexpression, while PHT1.12 (Glyma.20G021600) showed a more-than-10-fold downregulation (Figure 8A). The other genes, including the purple acid phosphatase gene *GmPAP27e* (Glyma.12G012000); the phosphate deficiency response transcription factor gene *GmPHR2* (Glyma.03G166400); the early nodulation genes *Nodulin2* (Glyma.13G114000), *Nodulin3* (Glyma.15G146000), *Nodulin4* (Glyma.17G117100), and *Nodulin5* (Glyma.17G117200); the WRKY transcription factor genes *WRKY1* (Glyma.04G061300), *WRKY3* (Glyma.17G035400), and *WRKY4* (Glyma.16G031900), the MYB transcription factor genes *MYB2* (Glyma.01G224900) and *MYB3* (Glyma.08G305000); the auxin efflux protein gene *PIN8b* (Glyma.17G057300); the GRAS protein essential for nod-factor signaling *NSP2b* (Glyma.06G216500); the auxin efflux protein gene *PIN9d*; the sucrose synthase gene *GmENOD100*; the small heat shock protein genes *GmHSP17.1* and *GmHSP17.9*; and the SPX domain protein gene *GmSPX5* were all upregulated via the overexpression of *GmAGL82* (Figure 8A).

Similarly, 20 genes exhibiting increases in expression because of *GmAGL82* overexpression were chosen for qRT-PCR nodule analysis (Figure 8B). Among these, the expression levels of three nodulation genes, *Nodulin1* (Glyma.08G120200), *Nodulin3* (Glyma.15G146000), not to mention the previously reported *NSP1a* (Glyma.07G039400), increased by 10.11%, 131%, and 116%, respectively (Figure 8B). The expression levels of two WRKY transcription factor genes, *WRKY1* (Glyma.04G061300) and *WRKY2* (Glyma.16G031900), increased by 85% and 88%, respectively (Figure 8A). The expression levels of two MYB transcription factor genes, *MYB1* (Glyma.01G224900) and *MYB2* (Glyma.12G199600), increased 4.5-fold and 2.2-fold, respectively. The other genes like ethylene response transcription factor genes, *ETR1* (Glyma.06G148400) and *ETR2* (Glyma.20G168500); phospholipase genes, *PHase1* (Glyma.02G142200), *PHase2* (Glyma.03G159000), and *PHase3* (Glyma.06G020500); trehalose-6-phosphate phosphatase genes, *ALT1* (Glyma.04G180900) and *ALT2* (Glyma.17G067800); the GDSL esterase gene (Glyma.13G231800); transcription factor genes *MADS* (Glyma.06G095700) and NAC181 (Glyma.19G108800); the fructose-2, 6-bisphosphatase gene *FRU* (Glyma.08G182300); as well as the small heat shock protein genes, *GmHSP17.1* and *GmHSP17.9*, were all upregulated by overexpressing *GmAGL82* in nodules (Figure 8B).

## 3. Discussion

Symbiotic nodulation is an important characteristic of leguminous crops and possesses great significance for legume growth. In recent years, several transcription factors have been discovered to participate in the nodulation of legumes, such as *Lotus japonicas* MYB transcription factor LjIPN2 [68], GRAS family transcription factors MtNSP1 and MtNSP2 in *Medicago truncatula* [69], and MADS transcription factor *GmNMHC5* in soybean [65]. The symbiotic nodulation process can be affected by phosphate deficiency stress [70,71]. However, the phosphate-starvation-responsive transcription factors functioning in symbiotic nodulation regulation remain largely unknown. In this study, we identified *GmAGL82* as an important regulator in soybean root nodulation in response to phosphate deficiency.

The MADS transcription factor family has received widespread recognition for its pivotal role in governing various plant growth and developmental processes through regulatory mechanisms, and it is also involved in legume nodulation. The MADS gene *NMH7* was initially discovered in *Medicago sativa* and confirmed to participate in the signaling transduction of rhizobial infection [72,73]. Research into soybeans has unveiled that the MADS gene *GmNMHC5* acts as a stimulatory controller of root growth and nodulation processes, which can interact with GmGAI to promote nodulation [65,74]. Here, a novel MADS family member, *GmAGL82*, was identified as being preferably expressed in nodules (Figure 2B), and the overexpression of *GmAGL82* significantly increased the nodule number and nodule fresh weight (Figure 4C,D). These results suggest that *GmAGL82* may positively regulate soybean root nodulation. To explore the underlying molecular mechanisms of *GmAGL82*, transcriptomic sequencing analysis was conducted on both roots and nodules of the *GmAGL82* overexpression line and the control line. From the results, we identified two small heat shock proteins, GmHSP17.1 and GmHSP17.9, which seem to be upregulated via the overexpression of *GmAGL82* in both roots and nodules (Table 1). It has been reported that GmHSP17.1 plays a positive role in soybean nodulation by interacting with GmRIP1, a peroxidase [75]. *GmAGL82* belongs to the MADS family of transcription factors and regulates downstream gene expression through binding to the CArG-box [CC(A/T)6GG] element present in the promoter region of target genes [76]. A further investigation of the promoter sequences indicated that the *GmHSP17.1* promoter contains two CArG elements, this suggests that *GmAGL82* may directly regulate the expression of the *GmHSP17.1* gene. *GmHSP17.9* has also been reported to perform a positive role in nodule development via interaction with *GmNOD100*, a sucrose essential for soybean nodulation [3]. Our results showed that overexpressing *GmAGL82* increased the expressions of both *GmHSP17.9* and *GmNOD100* (Table 1), while no CArG-box elements were found in their promoters. Research conducted on soybeans has unveiled MADS member GmNMHC5 to promote nodulation through interacting with GmGAI [65,74]. This suggests that there are other critical interacting partners for *GmAGL82* in the task of regulating soybean nodulation.

Beyond *GmHSP17.1*/*17.9* and *GmNOD100*, multiple crucial genes that function in soybean nodulation, growth, and development have been reported, including the auxin transport-related GmPIN9d, the SPX domain-containing protein GmSPX5, the NAC transcription factor GmNAC181, and the GRAS transcription factor GmNSP1a/2b (Table 1). GmPIN9d has been reported to act together with GmPIN1 later in nodule development, fine-tuning the auxin supply for nodule enlargement [77]. The overexpression of *GmAGL82* upregulated the expression of GmPIN9d in roots, indicating that *GmAGL82* may function as a mediator of auxin and thus regulate nodule development. Recent research showed that an SPX domain-containing protein, GmSPX5, can interact with GmNF-YC4 to regulate the asparagine synthetase-related gene GmASL6 for the purpose of mediating soybean nodule development [70]. Additionally, overexpression of *GmSPX5* promotes soybean nodule growth and development [70]. In this study, *GmSPX5* was significantly upregulated in overexpressing *GmAGL82* roots, indicating that *GmAGL82* may be involved in nodule development and asparagine metabolism through regulating GmSPX5. The GmNAC181 transcription factor, upregulated in nodules by overexpressing *GmAGL82*, was known as an important transcriptional activator of *GmNINa* in the nodulation pathway [78]. The overexpression of GmNAC181 promotes soybean nodule formation and increases the nodules number of soybean hairy root [78]. Additionally, *GmNSP1a* and *GmNSP2b* have been reported to be important regulators of early nodule formation [79,80]. In our results, *GmNSP1a* and *GmNSP2b* were upregulated by overexpressing *GmAGL82* in nodules and roots, respectively (Table 1), suggesting that *GmAGL82* may promote soybean nodulation through *GmNSP1a* and *GmNSP2b.*

Interestingly, it was found that the expression of *GmAGL82* was strongly induced by low-phosphate stress during the growth and development of nodules (Figure 2A). The overexpression of *GmAGL82* increased the nodule soluble phosphate concentration but decreased the root soluble phosphate concentration (Figure 4H). It is implied that *GmAGL82* may play an important role in low-phosphorus stress adaption through phosphate homeostasis regulation. Research has indicated that legume plant nodules have evolved adaptive strategies to cope with phosphorus deficiency; for instance, the plant is capable of prioritizing the transfer of phosphorus from other organs to the nodules in order to sustain higher levels of phosphorus [81,82]. In addition, it may enhance phosphorus uptake by utilizing internal phosphorus more effectively [27,83]. It is readily apparent that a heightened stimulation of nodule growth, augmentation of the nitrogen fixation capacity, and increase in soybean yield can be observed when soybeans overexpress the high-affinity phosphate transporters GmPT5 and GmPT7 [84]. Conversely, the activation of purple acid phosphatases serves as a strategic adaptation for nodules to acquire extra phosphorus. The elevated expression of *GmPAP12* noticeably enhances the quantity of nodules, their fresh weight, and the activity of nitrogenase when subjected to low-phosphorus stress [85]. In this study, several phosphate transporter- and purple acid phosphatase-related DEGs, including *GmPT11*, *GmPHT1.12/3.1/4.7*, *GmPHO1.12*, and *GmPAP27e*, were identified as being up- or downregulated in overexpressing *GmAGL82* roots or nodules (Table 1). The transcription factor GmWRKY45 has been reported to help plant tolerance to phosphate starvation, and overexpressing GmWRKY45 in Arabidopsis enhanced plant phosphorus concentration and upregulated several phosphate homeostasis-related genes, such as AtSPX1, AtPHO1, AtPH1;1/1;4/1;5, and AtACP5, under normal conditions [86]. Our results showed that overexpressing *GmAGL82* increased the expression of *GmWRKY45* in nodules, suggesting that *GmAGL82* may alter the phosphate homeostasis of nodules via GmWRKY45.

In addition to its high expression in nodules, *GmAGL82* also showed significant expression in soybean flowers (Figure 2B). Studies have reported that *OsMADS14*, *OsMADS15*, *OsMADS50*, and *OsMADS51* influence the flowering time in rice [87,88,89]. Furthermore, the research corpus indicates that *GmGAL1*, a homolog of *Arabidopsis thaliana* AtAGL20, enhances flowering under long-day conditions, as established by *Arabidopsis thaliana* transformation experiments [90]. The overexpression of *GmAGL1* not only promotes early maturity but also stimulates flowering and affects petal development [66]. By introducing *GmMADS28* into tobacco, researchers were able to induce early flowering and regulate floral organ number, fiber length, and partitioning [91]. These studies highlight the significance of MADS genes in regulating the formation of plant floral organs and the timing of flowering. These results suggest that *GmAGL82* may exhibit a dual regulatory role in coordinating soybean flowering and nodulation, offering a new research direction for investigating the coordinated regulation of these two important physiological processes.

## 4. Materials and Methods

### 4.1. Plant Materials and Growing Conditions

The phosphorus-efficient variety YC03-3, which was selected from the Root Biology Research Center of South China Agricultural University, was used as a soybean plant material. It had previously undergone characterization and utilization in other studies [35,92,93]. Well-adapted to phosphorus-deficient soils in the southern region of China, YC03-3 is a phosphorus-efficient soybean variety with a strong capacity for mobilizing and utilizing phosphorus [94,95]. The rhizobium strain BXYD3 was donated by the Root Biology Research Center of South China Agricultural University. Agrobacterium tumefaciens GV3101 and Agrobacterium tumefaciens K599 were provided by Shanghai Weidi Biological Company. The rhizobium strain used in the experiments was BXYD3. Initially, the soybean seeds were sown in sandy soil to facilitate germination. After five days, uniform and healthy seedlings were selected and inoculated with rhizobia (OD600 = 1.0). Subsequently, seedlings were transferred to a modified nutrient solution [70] for further cultivation. The treatment involved a normal phosphorus supply (+P) with a concentration of 250 μmol/L KH_2_PO_4_, while the experimental treatment involved a low phosphorus supply (−P) with a concentration of 5 μmol/L KH_2_PO_4_. Specifically, the base nutrient solution contained KNO_3_ 30.33 mg/L, CaCl_2_ 133.18 mg/L, MgCl_2_ 5.08 mg/L, K_2_SO_4_ 156.843 mg/L, MgSO_4_∙7H_2_O 123.24 mg/L, MnSO_4_∙H_2_O 0.254 mg/L, ZnSO_4_∙7H_2_O 0.431 mg/L, CuSO_4_∙5H_2_O 0.125 mg/L, (NH)_2_MoO_24_∙4H_2_O 0.2 mg/L, Fe-EDTA(Na) 14.68 mg/L, NaB_4_O_7_∙10H_2_O 0.95 mg/L. The nutrient solution was changed every five days. We maintained a pH range of 5.8–6.0 in the nutrient solution by adjusting it with 1 M KOH or H_2_SO_4_ (Kermel, Tianjin, China). Samples were collected at different rhizobia growth periods by harvesting soybean rhizomes subjected to various phosphorus treatments at 15, 20, 25, 40, 50, and 60 days after planting. Additionally, samples were collected from different tissue sites including roots, stems, leaves, flowers, fruit pods, seeds, and nodules on day 32 of the culture when the soybeans were flowering. Employing liquid nitrogen as a rapid freezing agent, the collected samples were swiftly subjected to cryopreservation, achieving a low temperature of −80 °C, with the express aim of facilitating seamless RNA extraction. Samples from different plant tissues were collected with an appropriate amount of plant tissue, ground into a fine powder with liquid nitrogen and 100–200 mg of fine powder was taken for RNA extraction. The extracted total RNA was then assessed for its concentration, aiming for a range of 300–400 μg/mL, and purity, aiming for a range of 1.7–2.0., it was reverse transcribed into cDNA. Subsequent real-time fluorescence quantitative experiments included three biological replicates. For the rhizobium inoculation process. Initially, rhizobium strain BXYD3 was activated on agar plates, and a single clone was selected for inoculation into YMA culture medium (1000 mL contained 10 g mannitol, 0.2 g MgSO_4_·7H_2_O, 0.1 g NaCl, 3 g yeast extract, 0.25 g K_2_HPO_4_, 0.25 g KH_2_PO_4_, 0.05 g CaCl_2_). Incubate at 28 °C for about 4 days until OD_600_ is about 1.0. Subsequently, the transgenic soybean hair roots were immersed in rhizobial suspension for 1 h, and then transplanted into soybean base nutrient solution containing different phosphorus concentrations for culture, and a ventilation device was installed with 15 min of air exchange per hour. The harvest was harvested after 28 days of treatment, and the corresponding indexes were determined. During this period, the nutrient solution is changed once a week, and the pH is controlled between 5.8 and 6.0 every day. Soybean plants are grown in a solar greenhouse with an average daytime temperature of 30 °C, an average night temperature of 25 °C, and air humidity of 75%. For artificial lighting, T8-integrated LED tubes with a power rating of 18W are used. The average daily photoperiod is 13 h, providing an average light intensity of 800 μmol·m^−2^·s^−1^.

### 4.2. RNA Extraction and Quantitative Real-Time Polymerase Chain Reaction Testing

RNA extraction kits (Yeasen, Shanghai, China) were used to isolate total RNA from the nodules and roots of soybean plants. Genomic DNA (gDNA) was eliminated, and cDNA was generated by employing PrimeScript RT reagent kits (Takara, Maebashi, Japan). The real-time fluorescence quantitative PCR instrument received from Bio-Rad (Hercules, CA, USA) was utilized to conduct the qRT-PCR analysis, following the procedures delineated in a prior publication [11]. In short, the cDNA sample to be tested was diluted 10 times as the template, and the program comprisedcc an initial denaturation step at 95 °C for 30 s, which was followed by 40 cycles of amplification, each involving a denaturation step for 5 s at 95 °C, annealing at 58 °C for 1 min, and extension at 72 °C for 30 s. To serve as a control, the internal reference gene *GmEF1-α* (Glyma17g23900) was used to assess the expression levels of the genes of interest, which were normalized to the internal control genes. This was achieved by calculating the transcript ratios between them, working in accordance with the methodology in a previous study [96]. Appendix A lists the primers that were utilized in qRT-PCR testing. Three biological replications were included.

### 4.3. Determination of Total Phosphorus Content and Soluble Phosphate Concentration

The analysis of total phosphorus (P) and soluble phosphate (Pi) concentrations in the soybean samples was conducted meticulously, and we adhered to a previously established protocol [4]. Briefly, the soybean roots and nodules were harvested, dried, and crushed. About 0.1000 g of sample was digested using H_2_SO_4_ and H_2_O_2_, a process which continued until the sample solution became clear or colorless. The total P concentrations were determined using a colorimetric method, as described previously [97]. Specifically, to determine the total phosphorus content, an appropriate amount of the solution was taken and transferred into a stoppered test tube with a capacity of 20 mL. A 20 µL volume of dinitrophenol indicator (Macklin, Shanghai, China) was added, and the final solution volume was brought up to approximately 3 mL using double-distilled water. The solution was adjusted to a slight yellow color using 2 M NaOH (Kermel, Tianjin, China). Subsequently, 0.5 mL of molybdenum–antimony anti-color developer was added, and the total solution volume was adjusted to approximately 5 mL using double-distilled water. We performed the reaction for 30 min at room temperature. For the colorimetric analysis, the absorbance values were measured using a quartz cuvette (Allrenta, Beijing, China) and a UV–Vis spectrophotometer (Shanghai Yuanxi UV-5100B, Shanghai, China) at a wavelength of 700 nm. The total phosphorus content was expressed as the phosphorus concentration multiplied by the total dry weight.

To determine the soluble phosphorus concentration, we weighed 0.1000 g of fresh sample from each part of the roots and nodules. The samples were then extracted with 600 µL of double-distilled water and transferred to 1.5 mL centrifuge tubes. The bowls used for extraction were rinsed with an additional 600 µL of double-distilled water, which was also transferred to the respective centrifuge tubes. After centrifugation at 12,000 rpm for 30 min using an Eppendorf 5415D centrifuge (Hamburg, Germany), the supernatant was isolated from the tubes into pristine 1.5 mL centrifuge tubes in preparation for subsequent quantification. For the measurement, a 20 mL stoppered tube was filled with the obtained supernatant sample, and 500 µL of the molybdenum–antimony anti-colorant, and the final solution volume was adjusted to 5 mL using double-distilled water. The contents were thoroughly mixed. To determine the soluble phosphorus concentration in the root system and nodules, a calibration curve was constructed using soluble phosphorus concentration standards.

### 4.4. Determination of Total Nitrogen Content

The nitrogen content of each fraction was determined using a fully automated Kjeltec nitrogen tester (FOSS Kjelte 8400, Denmark, Europe). To establish the concentrations of *N*, approximately 0.1 g (dry weight) of plant materials was subjected to digestion in H_2_SO_4_ prior to the measurement of total *N* concentrations, working in accordance with a previously described method [98]. The nitrogen concentration in the sample was calculated using the formula $N(mg/g)=\frac{(T-B)\times N\times 14.007\times V1}{V2\times W}$, where *T* represents the volume (in mL) of hydrochloric acid consumed during the titration of the sample; *B* is the volume (in mL) of hydrochloric acid consumed during the blank experiment; *N* denotes the molarity (0.1115 mol/L) of the hydrochloric acid; *V*1 represents the total volume (in mL) of the cooking solution; *V*2 is the volume (in mL) of the sample used for the analysis; and *W* denotes the weight (in g) of the dry sample utilized during the cooking process. The final total nitrogen content was expressed as the product of the nitrogen concentration and the total dry weight of the sample.

### 4.5. Analysis of the Subcellular Localization of GmAGL82

The subcellular localization analysis of *GmAGL82* was examined via its transient expression in the epidermal cells of tobacco (*Nicotiana benthamiana*) leaves, as described previously [99]. The open reading frame (ORF) of *GmAGL82*, which lacks a stop codon, was amplified and subsequently inserted into the pBWA(V)HS-Glosgfp vector using specific primers (Appendix A). *Agrobacterium tumefaciens* GV3101 (Weidibio, Shanghai, China) containing the pBWA(V)Hs-*GmAGL82* plasmid was activated and expanded. Subsequently, the bacterial cultures underwent centrifugation at 5000 revolutions per minute (rpm) for a duration of 10 min. Following this, the resulting pellet was resuspended in a solution at an optical density (OD) of 1.0. An infiltrating solution, consisting of 10 mM MgCl_2_, 10 mM MES, and 100 µM acetosyringone, was prepared and incubated at 22–24 °C for 3 h in the dark. The bacterial solution was then injected through a syringe into the leaf epidermis of tobacco that had been growing for 3–4 weeks. After 3 days of transformation, laser confocal scanning microscopy (LsM780, Zeiss, Jena, Germany) was employed to conduct fluorescence imaging. Images were captured using an excitation wavelength of 488 nm and an emission wavelength of 507 nm. The empty vector served as the control in this experiment.

### 4.6. Exploring GmAGL82 through Bioinformatics and Phylogenetic Tree Analysis

By exploring the Phytozome database (https://phytozome-next.jgi.doe.gov/ (accessed on 17 April 2023) with the sequence ID Glyma.19G045900, we were able to retrieve information about the *GmAGL82* gene, including its ORF length, number of exons and introns, and amino acid length. The prediction of conserved structural domains of the GmAGL82 protein was performed using the NCBI conserved domain website (https://www.ncbi.nlm.nih.gov/Structure/cdd/wrpsb.cgi (accessed on 19 April 2023). In order to predict the cis-acting elements present in the promoter region of the *GmAGL82* gene, we made use of the Plant CARE website, which can be accessed at http://bioinformatics.psb.ugent.be/webtools/plantcare/html/ (accessed on 21 April 2023). The region under analysis encompassed a stretch of 2000 base pairs located upstream of the 5′ UTR sequence. The phylogenetic tree analysis was conducted using MEGA v. 6.0 software. We utilized the neighbor-joining method and conducted bootstrap statistical testing with a total of 1000 replicates.

### 4.7. GmAGL82 Overexpression in Soybean Hairy Roots and Nodules

The open reading frame (ORF) of *GmAGL82* was amplified and subsequently inserted into the pTF101s vector using specific primers (Appendix A). The *GmAGL82*-pTF101s (OX) or empty-vector (CK) construct was introduced into *Agrobacterium tumefaciens* K599 to infect soybean seedlings to generate transgenic hairy roots, following a previously described method [37]. Specifically, after seed germination, an appropriate amount of the *Agrobacterium tumefaciens* K599, carrying *GmAGL82*-pTF101s or pTF101s empty vectors, was taken using a 1 mL syringe. The needle was then dipped and coated at a position 0.5 mm below the cotyledon node of the soybean seedlings to create a perforation for the application of bacterial cells, thereby increasing the infected area. After approximately 10 days, the visible growth of hairy roots could be observed. Once these roots reached a length of approximately 10 cm, the original primary root was removed, leaving behind only the transgenic hairy roots, thus resulting in the generation of transgenic composite plants. After inoculation with rhizobia, the plants were transferred to a base nutrient solution. After 28 days, the roots and nodules were harvested. The expression levels of the transgenic hairy roots and nodules were determined using RT-qPCR, and those with desirable expression levels were selected for inclusion in the subsequent analyses.

### 4.8. Creation of cDNA Libraries and Analysis of Transcriptomic Sequencing

Root and nodule samples were collected separately for total RNA extraction and mRNA library preparation. The samples included control (transgenic composite plant lines transformed with an empty vector) and treatment groups (transgenic composite plants overexpressing *GmAGL82*). Transcriptomic sequencing analysis was conducted following a previously described protocol [71]. Each sample was analyzed using three biological replicates. In brief, high-quality RNA was extracted from the samples. For the purification of eukaryotic mRNA, we opted to use magnetic beads coated with oligo (dT). Subsequently, the purified mRNA was subjected to random fragmentation, after which it was transformed into cDNA through the process of reverse transcription using random primers. Afterwards, the double-stranded cDNA obtained was purified, followed by end-repair and the selection of fragments according to their size. The PCR enrichment process was employed to obtain the final cDNA library. To ensure the integrity of the constructed libraries, we assessed the effective concentration and insert length. The raw sequencing data (Raw Data) underwent filtering to remove reads with low quality (average base quality value less than 20) and a number of ambiguous bases (N) greater than five. The clean reads were aligned against a reference genome, using specialized software to determine their positions. We used the FeatureCounts software (Release 2.0.3) to calculate the FPKM values of each gene in each sample, which helped us to determine their respective levels of gene expression. Differential expression analysis was performed using DEseq2, with criteria for differential significance set as padj < 0.05 and |log2fold change| > 1. We then carried out GO functional enrichment analysis and KEGG pathway enrichment analysis. In order to identify terms that exhibited notable enrichment, we utilized Fisher’s Exact Test and applied a *p*-value threshold of <0.05 after carrying out adjustments.

### 4.9. Statistical Analysis

The execution of statistical analysis in this study entailed utilizing the Microsoft Excel 2021 software received from Microsoft Company, United States, and the SPSS program (v25.0) from SPSS Institute, United States. Throughout the analyses, we calculated the mean and standard error, while the assessment of group differences was conducted through the application of the Student’s *t*-test. The graphs were generated using Origin (Version Number ”Version 2021”, OriginLab Corporation, Northampton, MA, USA).

## 5. Conclusions

In this study, the MADS-box gene *GmAGL82* was overexpressed in soybeans. Phenotypic analysis revealed that the overexpression of *GmAGL82* resulted not only in increased biomass, particularly in terms of nodule quantity, but also in enhanced nitrogen and phosphorus content, which influenced the soluble phosphorus concentration. Moreover, the transcript levels of *GmAGL82* were found to be elevated in mature nodules exposed to low-phosphorus conditions. This contrasts their response contrast to conditions with high levels of phosphorus. The transcriptomic sequencing analysis of *GmAGL82*-overexpressing plants suggested that *GmAGL82* might facilitate nodule growth and development by regulating the genes associated with soybean nodulation. This finding also implied the involvement of other interacting proteins that cooperate with *GmAGL82* to regulate additional genes.

## Figures and Tables

**Figure 1 ijms-25-01802-f001:**
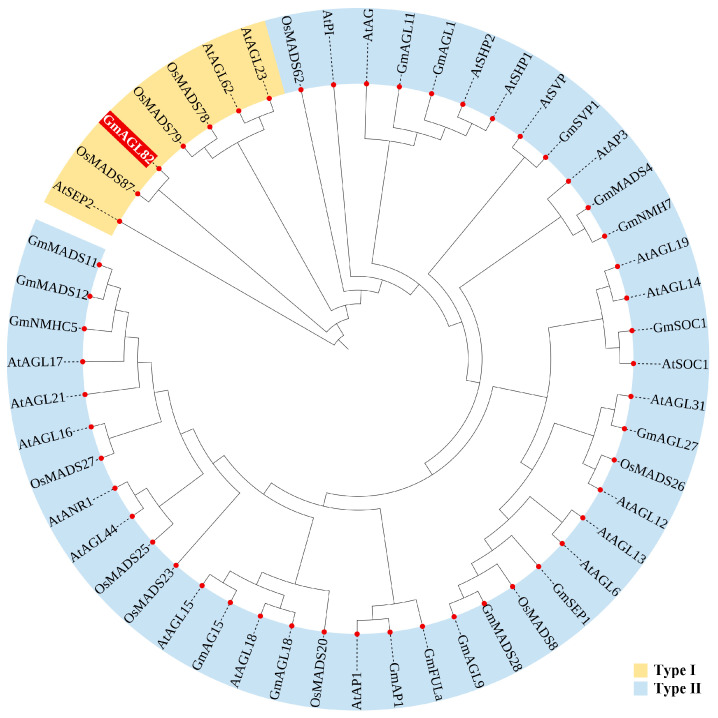
Phylogenetic tree analysis of MADS family proteins. At: representative *Arabidopsis thaliana*; Os: representative *Oryza sativa*; Gm: representative *Glycine max*. The red shading highlights the gene of interest in this study A phylogenetic tree was constructed using MEGA v. 6.0 software using the neighbor-joining method. Amino acid sequences and corresponding identification numbers of the MADS-box family were obtained from the National Center for Biotechnology Information (https://www.ncbi.nlm.nih.gov/ (accessed on 17 April 2023) and the Phytozome (http://phytozome.net/ accessed on 17 April 2023) database: AtAP1 (AT1G69120), AtAG (AT4G18960), AtSOC1 (AT2G45660), AtANR1 (AT2G14210), AtPI (AT5G20240), AtAP3 (AT3G54340), AtAGL6 (NP_182089.1), AtAGL12 (AT1G71692), AtAGL13 (AAC49081), AtAGL14 (AT4G11880), AtAGL15 (AT5G13790), AtAGL16 (AT3G57230), AtAGL17 (AT2G22630), AtAGL18 (AT3G57390), AtAGL19 (AAG37901), AtAGL21 (AT4G37940), AtAGL23 (AT1G65360), AtAGL31 (NP001119498.1), AtAGL44 (NP_179033.1), AtAGL62 (AT5G60440), AtSHP1 (OAP06129), AtSEP2 (AAF61626), AtSHP2 (NP_850377), AtSVP (OAP09056), OsMADS8 (Q9SAR1), OsMADS20 (LOC_Os12g31748), OsMADS23 (LOC_Os08g33488), OsMADS25 (LOC_Os04g23910), OsMADS26 (LOC_Os08g02070), OsMADS27 (LOC_Os02g36924), OsMADS62 (LOC_Os08g38590), OsMADS78 (LOC_Os09g02830), OsMADS79 (LOC_Os01g74440), OsMADS87 (LOC_Os03g38610), GmNMHC5 (Glyma13g32810), GmNMH7 (Glyma06g02990), GmSVP (Glyma.01G023500), GmSEP1 (Glyma19g04320), GmSOC1 (NP_001236377), GmAGL1 (Glyma.14G027251), GmAGL9 (ACA24481.1), GmAGL11 (Glyma06g48270), GmAG15 (Glyma.11G158812), GmAGL18 (XP_006575259), GmAGL27 (NP_177833.3), GmMADS4 (Glyma01g37470), GmMADSI1 (XP_014623536), GmMADSI2 (XP_025981482), GmMADS28 (NP001236390.1), GmAP1 (XP_003547792), GmFULa (ahi43155), and GmAGL82 (Glyma.19G045900).

**Figure 2 ijms-25-01802-f002:**
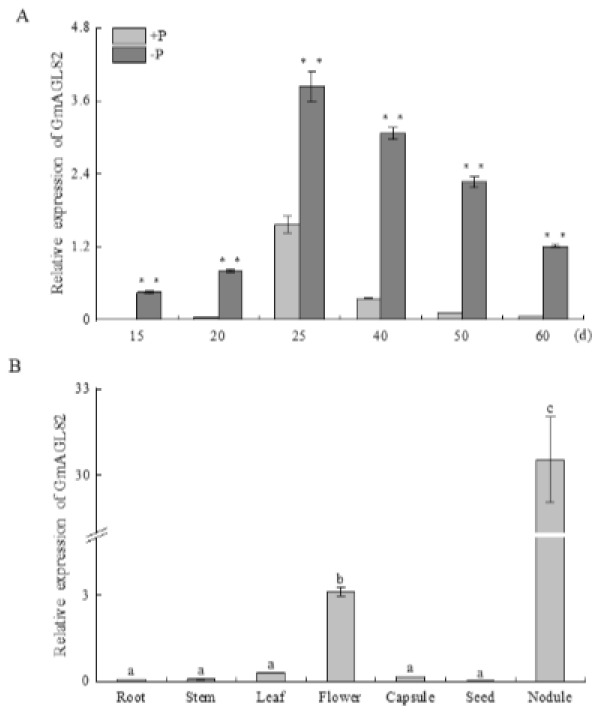
*GmAGL82* expression pattern. (**A**) The expression of *GmAGL82* in root nodules at different time points. (**B**) Expression analysis of *GmAGL82* in different parts of the soybean plant. Experimental data are expressed as the mean and standard error (SE) of three biological replicates. Significance analysis was performed using the Waller–Duncan model. Different letters indicate significant differences between the different concentrations in different tissues at the *p* < 0.05 level. Asterisks indicate significant differences (**, *p* < 0.01).

**Figure 3 ijms-25-01802-f003:**
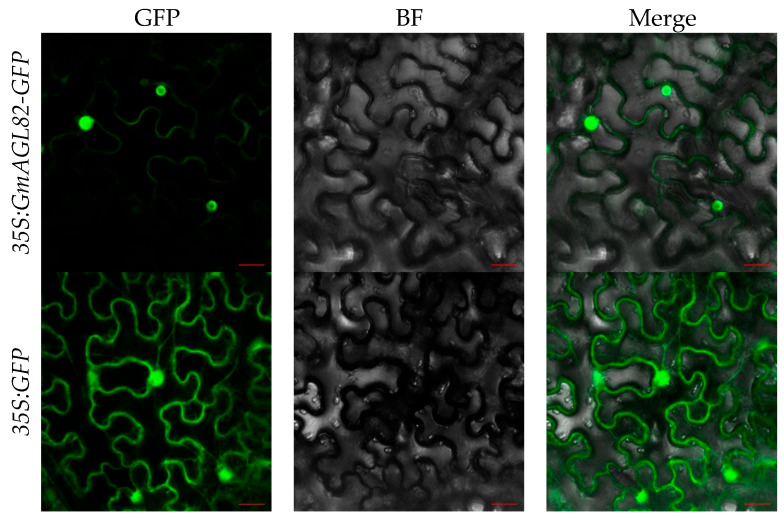
Subcellular localization of GmAGL82. GFP: green fluorescent protein channel; BF: bright field; Merge: fusion. The scale bar is 20 μm.

**Figure 4 ijms-25-01802-f004:**
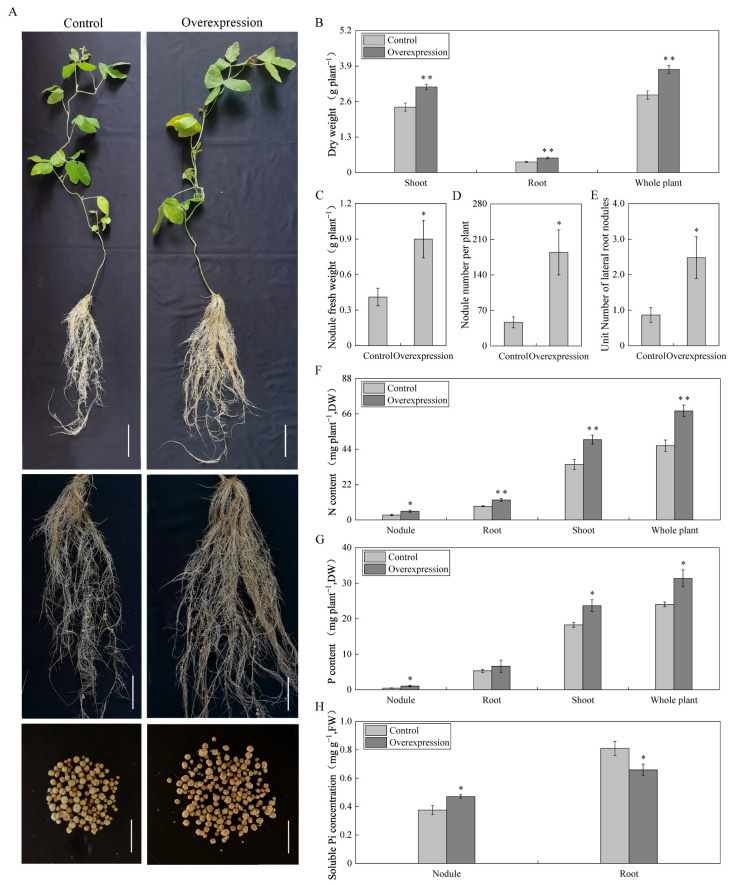
Effects of overexpressing *GmAGL82* on the growth of soybean root nodules. Control: transgenic composite plants transformed using empty vectors. Overexpression: transgenic composite plants overexpressing *GmAGL82*. (**A**) Plant phenotypes 28 days after inoculation with rhizobia, where the whole-plant scale is 10 cm, the hairy root, and rhizome symbiosis phenotype scale is 5 cm, and the summary tumor phenotype scale is 1 cm; (**B**) dry weight of shoots, roots, and whole plant; (**C**) nodule fresh weight; (**D**) number of nodules per plant; (**E**) number of root nodules per unit of root length. Effect of overexpression of *GmAGL82* on the (**F**) nitrogen content, (**G**) phosphorus content, and (**H**) soluble phosphorus concentration in soybean composite plants. Data in the graph are shown as the means and standard errors of five biological replicates; * indicates a significant difference (*p* < 0.05); and ** indicates a highly significant difference between control and overexpression plants (*p* < 0.01).

**Figure 5 ijms-25-01802-f005:**
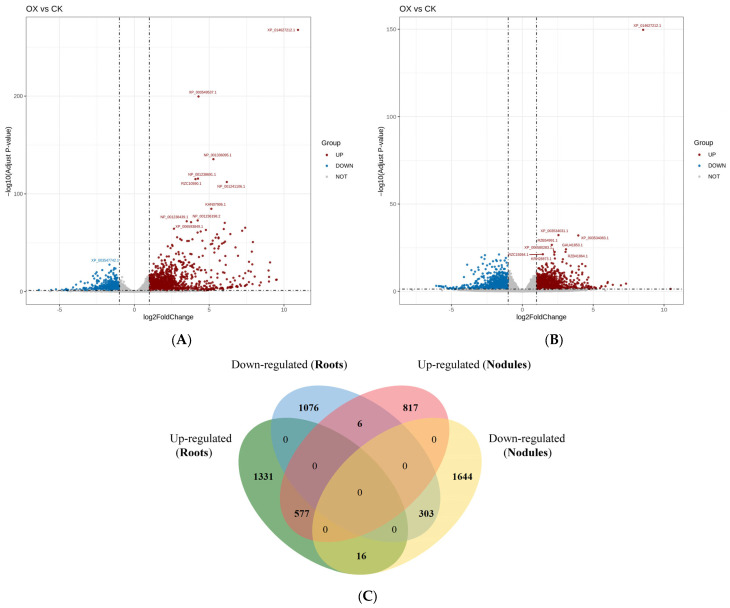
Volcano plots and a Venn diagram of the differential gene expression distribution. (**A**) Volcano plot of the differential gene expression distribution in CK and OX roots. (**B**) Volcano plot of the differential gene expression distribution between CK and OX nodules. In (**A**,**B**), the horizontal coordinates indicate gene expression changes in the different experimental groups or samples. The vertical coordinates indicate the statistical significance of the changes in gene expression. The scattered dots in the graph represent individual genes, with gray dots indicating genes with no significant differences, red dots indicating upregulated genes with significant differences, and blue dots indicating downregulated genes with significant differences. (**C**) Genes that were differentially expressed between *GmAGL82*-overexpressing roots and nodules and control treatments are expressed in a Venn diagram.

**Figure 6 ijms-25-01802-f006:**
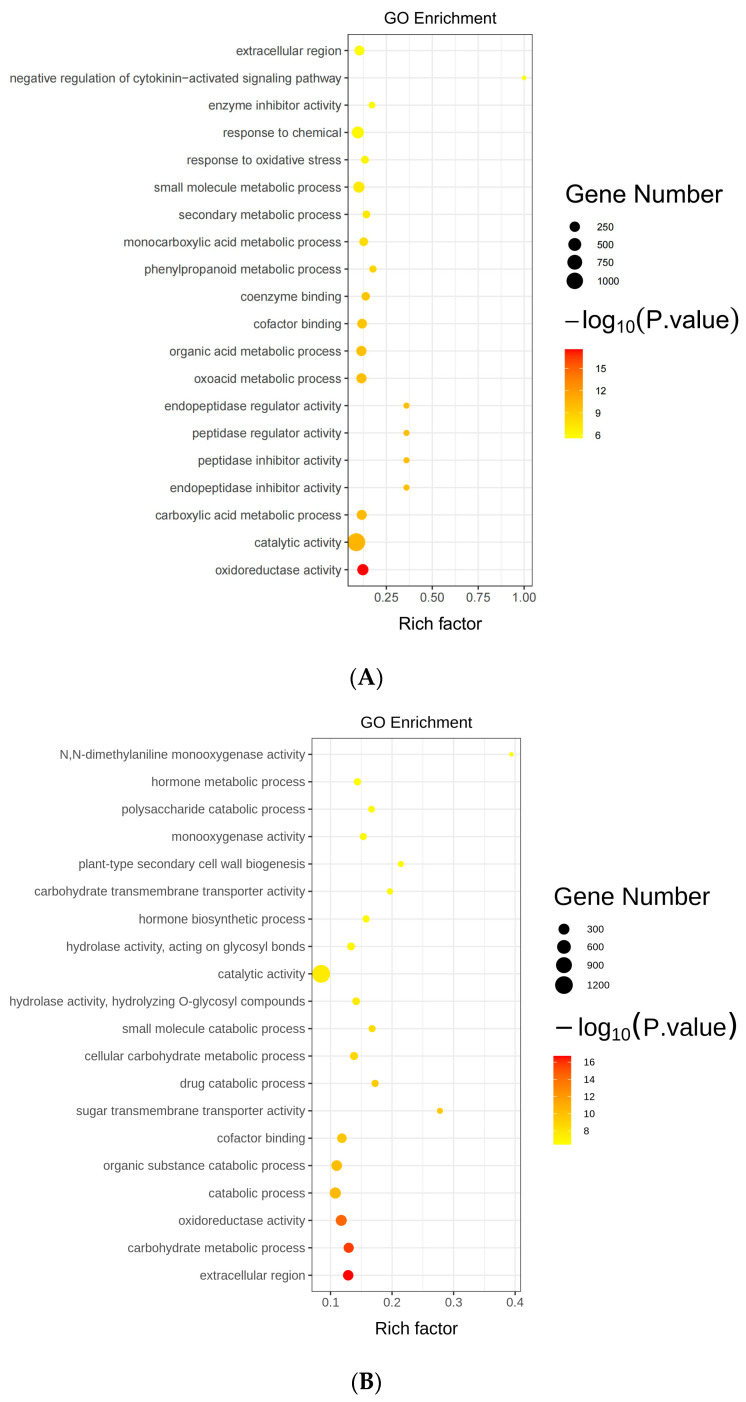
GO functional enrichment of differentially expressed genes (**A**) GO analysis of DEGs (*p*-value ≤ 0.05) in the roots (**B**) GO analysis of DEGs (*p*-value ≤ 0.05) in the nodules.

**Figure 7 ijms-25-01802-f007:**
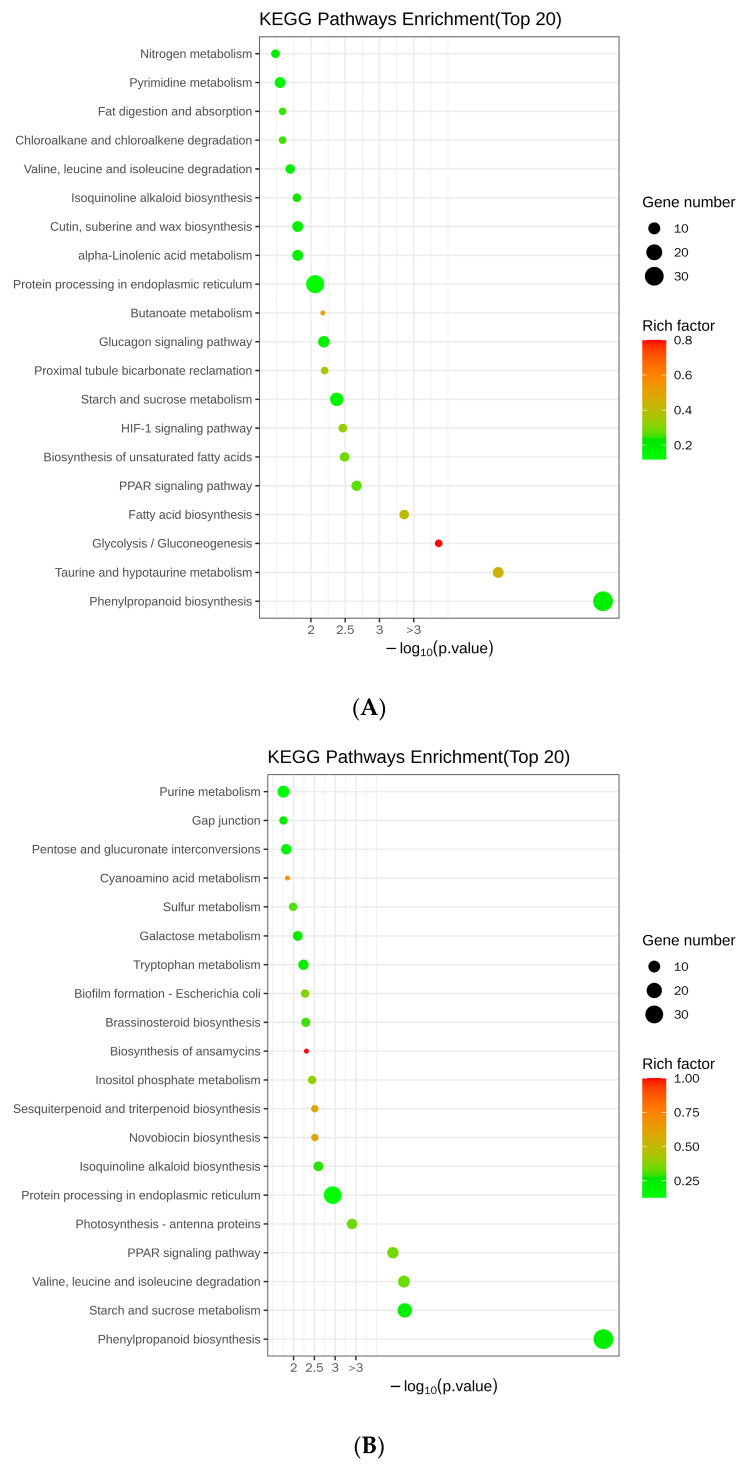
KEGG enrichment bubble map of DEGs. (**A**) KEGG analysis of DEGs in the roots. (**B**) KEGG analysis of DEGs in nodules. The vertical coordinates indicate the different metabolic pathways and the horizontal coordinates indicate the *p*-values corresponding to the metabolic pathways. The size of the enrichment factor is indicated by the color of the dot, where the larger the value is, the closer the color will be to red. The number of differential genes contained under each pathway is indicated by the size of the scatter.

**Figure 8 ijms-25-01802-f008:**
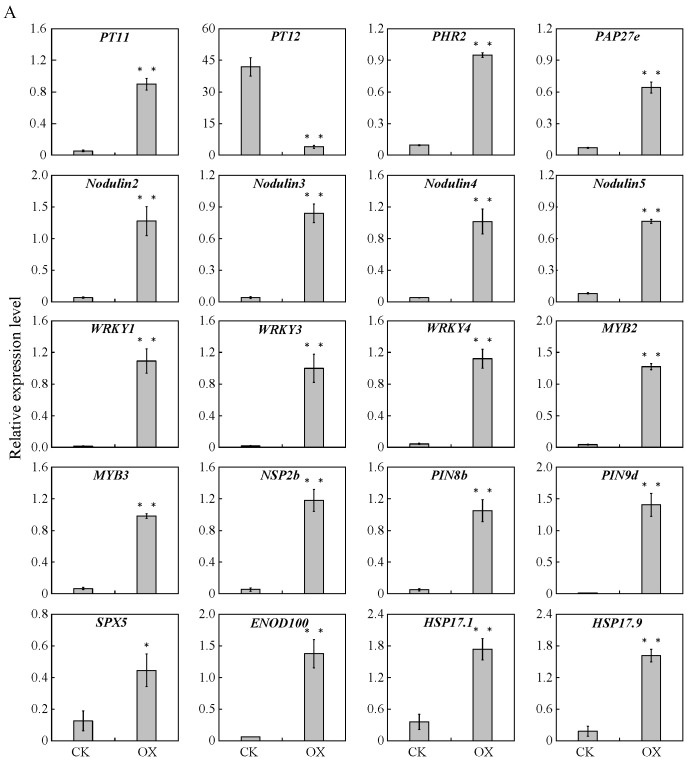
qRT-PCR analysis of differentially expressed genes in the transformed empty-vector control (CK) and *GmAGL82*-overexpressing (OX) (**A**) roots and (**B**) nodules. All data are shown as the means and standard errors of three replicates. * indicates a significant difference compared to the CK (Student’s *t*-test, *p* < 0.05). ** indicates a significant difference compared to the CK (Student’s *t*-test, *p* < 0.01). ALT, alpha-trehalose-phosphate synthase; bZIP, BZIP transcription factor; ENOD, hypothetical protein GLYMA; ETR, ethylene-responsive transcription factor; FRU, fructose-1; GDSL, GDSL esterase; HSP17, heat shock protein 17; MADS, MADS-box protein; MYB, MYB transcription factor; NAC181, transcriptional factor NAC11; Nodulin, early nodulin; NSP, nodulation signaling pathway protein 1; PAP, hypothetical protein GLYMA; PHase, phospholipase; PHR, protein phosphate starvation response; PIN, PIN auxin efflux transporter family protein; PT, inorganic phosphate transporter; SPX, SPX-domain-containing protein; WRKY, WRKY transcription factor.

**Table 1 ijms-25-01802-t001:** The DEGs involved in soybean nodulation and P signaling.

Gene Locus	Name	log_2_Fold Change		Description
		Roots	Nodules	
Glyma.06G157800	GmHSP17.1	5.86	2.36	Small heat shock protein
Glyma.04G054400	GmHSP17.9	6.42	3.21	Small heat shock protein
Glyma.17G045800	GmENOD100	1.58	-	Sucrose synthetase
Glyma.17G057300	GmPIN8b	1.15	-	Auxin transfer protein
Glyma.15G208600	GmPIN9d	3.22	-	Auxin transfer protein
Glyma.10G261900	GmSPX5	2.84	-	SPX domain protein
Glyma.19G164300	GmPT11	1.57	-	Phosphate transporter protein
Glyma.20G021600	GmPHT1.12	−1.16	-	Phosphate transporter protein
Glyma.01G157100	GmPHT3.1	−1.65	-	Phosphate transporter protein
Glyma.20G032500	GmPHO1.12	1.51	-	Phosphate transporter protein
Glyma.13G162900	GmPHT4.7	-	1.43	Phosphate transporter protein
Glyma.19G108800	GmNAC181	-	1.02	NAC protein family
Glyma.07G039400	GmNSP1a	-	1.21	GRAS transcription factor
Glyma.06G216500	GmNSP2b	2.06	-	GRAS transcription factor
Glyma.12G012000	GmPAP27e	1.98	-	Purple acid phosphatase
Glyma.03G166400	GmPHR2	1.35	-	PHR1 transcription factor
Glyma.03G220800	GmWRKY45	-	1.36	WRKY transcription factor

Note, “-”means no expression responses were found.

## Data Availability

The original contributions presented in the study are publicly available. This RAN-seq raw data can be found on the NCBI repository, accession number: PRJNA1005769.

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
