# Peer review of "Unregulated GmAGL82 due to Phosphorus Deficiency Positively Regulates Root Nodule Growth in Soybean"

_ijms, 2024, doi:10.3390/ijms25031802_

Round 1

Reviewer 1 Report

Comments and Suggestions for Authors

The article devoted on the overexpression of the MADS-box gene GmAGL82 in soybeans. Phenotypic analysis revealed that this overexpression not only significantly enhances biomass, particularly by increasing the number of nodules, but also raises the nitrogen and phosphorus contents. This, in turn, affects the concentration of soluble phosphorus. The methodologies employed are appropriate, and the conclusions align well with the stated objectives. However, there is a need for a more detailed description of the rhizobium inoculation process, including specifics on the medium, timing, and temperature. This section would benefit from an expanded and more detailed exposition. Furthermore, it is essential to provide a more comprehensive account of how the soybean plants were cultivated. Details such as whether they were grown in a greenhouse or a climate chamber, the specific type of lamps used if in a climate chamber, the intensity of light flow, and air humidity levels should be included. Additionally, it is imperative to specify more precisely the number of roots, stems, leaves, flowers, fruit pods, seeds, and nodules that were analyzed, along with the number of biological analytical replicates. I would recommend reconsidering the title to avoid abbreviations, although this decision ultimately rests with the authors.

Comments on the Quality of English Language

Moderate editing of English language required

Reviewer 2 Report

Comments and Suggestions for Authors

In the reviewed manuscript a MADS-box family member GmAGL82, preferentially expressed in nodules and whose expression is significantly elevated under phosphate deficiency was extensively studied in the phosphorus-efficient soybean variety YC03-3, both at the phenotypic and molecular levels. The research is relevant for understanding the regulatory mechanisms underlying nodule development in soybean plants. The data indicate a possible involvement of GmAGL82 in the regulation of a number of genes associated with nodule growth, and confirm its participation in the phosphorus signaling pathway and nodulation in soybean.

Comments:

1)         Please add information on the occurrence of acidic soils in the soybean growing areas.

2)         Please specify the term “phosphorus-efficient variety” and describe briefly the accession YC03-3 (origin, distinctive characteristics).

3)         Line 178 – Figure S1 is absent in the supplementary file

4)         Please check writing GmAGL82 throughout the text (italics for gene, upright for protein)

5)         Line 433 – Please correct: “Symbiotic nodulation is…

6)         Line 452 – Please correct: “underlying”

7)         Line 557 – Please change “peanut plants” to “soybean plants”.

Reviewer 3 Report

Comments and Suggestions for Authors

The paper describes a very interesting aspect of nodule formation in the roots of soybeans. This process is known to be induced by low phosphorus concentration, and the Authors demonstrated a positive role of one MADS transcription factor in such conditions of stress for roots.

Phosphorus is one of the important nutrients for plants, and its deficiency may negatively affect the plant's ability to maintain the nodulation process in which legumes (e.g. clover, peas, soybeans) create root nodules that contain bacteria of the Rhizobium genus. Appropriate availability of soluble organic phosphates is important for effective cooperation between the plant and nodule bacteria. In conditions of phosphorus deficiency, the nodulation process and the plant's ability to use nitrogen may be disturbed in the symbiosis associated with it. Since P availability is limiting in most of the world's soils, plants have evolved with a complex network of genes and their regulatory mechanisms to cope with soil P deficiency. Among them, purple acid phosphatases (PAPs) are predominantly associated with P remobilization within the plant and acquisition from the soil by hydrolyzing organic P compounds (Bhaduria & Giri, Plant Cell Rep. 2022). While new functions have been assigned to PAPs, the underlying mechanism of P uptake remains poorly understood. Maybe the GmAGL82 transcription factor can be involved in the regulation of the candidate gene of PAP.

From a molecular biology perspective, the regulation of root nodule growth in legumes is associated with many molecular processes, including the control of gene expression. Transcription factors from the MADS family (MCM1, Agamous, DEF, SRF) are known for their role in the regulation of flowering and flower development in plants, but their influence on the development of root nodules may be related to various aspects of root morphogenesis and interactions with nodule bacteria. The research of Song et al. presents an interesting demonstration of a MADS family member, a gene GmAGL82 (Glyma19G045900), that is upregulated in root nodules under low phosphate conditions. The in-planta study of MADS gene expression during various stages of nodule growth in soybean plants confirmed the role of this transcription factor in signaling pathways related to nodulation.

The text, the methodology, and the results are not fully comprehensive and need some clarification (details are given below). The discussion and conclusions do not raise any objections, they are justified by the obtained results. References are complete but need one unified layout.

Specific comments to improve the quality of the manuscript are given below:

lines 141, 294: Please provide a better quality graph to Figure 1 and Figure 5. In the description of Figure 1 you mention “MEGA v. 6.0” but in Materials and Methods “MEGA v. 5.05” – choose the correct one

line 168: Figure S1 was not uploaded to the Review Report Form, so I cannot see it.

lines 407- : It will be better to put the names and abbreviations with full names in alphabetic order.

line 535 (in Materials and Methods): add the origin of the rhizobium strains BXYD3 (company or own culture) as it was mentioned for Agrobacterium tumefaciens GV3101 in line 621. Check if all strains have their origin.

line 542: I did not really understand the use of the “low-nitrogen nutrient solution”. Please write all variants of the experiment with descriptions, e.g. variant 1) control plant tissues: 1.a) normal supply of P; 1.b.) normal supply of N, variant 2) plant tissues with lower supply of P, but normal supply in N, etc.

lines 727- 932: All references have redundant numbering. The bibliography needs a layout correction (missing dots at the end of titles. Dots are present / or absent at the end, etc.

line 903: Add the complete reference as it is in Google Scholar “Frontiers in Plant Science 2020, 11, 450.”

line 927: check this strange citation “(24 %@ 1615-9853)”, it should be “Proteomics, 11(24), 4648-4659.” Revise also lines 835, 840, 901, 931 etc.

lines 735, 764: unify the spelling of journal names, i.e. in these lines there are “Plant physiology” and “Plant Physiology”. Check all the journal names.

To sum up, I recommend publishing the article after those corrections. 

Round 2

Reviewer 3 Report

Comments and Suggestions for Authors

The Authors have carefully revised and improved their manuscript. All my remarks have been taken into account and I have no more scientific remarks. One newly added sentence in the conclusion remains unclear to me, e.g. "This contrasts their response contrast to conditions with high levels of phosphorus" (lines 762-763), and needs to be rephrased.